# Micropulse Laser Treatment of Retinal Diseases

**DOI:** 10.3390/jcm8020242

**Published:** 2019-02-13

**Authors:** Maciej Gawęcki

**Affiliations:** Dobry Wzrok Ophthalmological Clinic, Kliniczna 1B/2, 80-402 Gdansk, Poland; gawecki@use.pl; Tel.: +48-501788654

**Keywords:** subthreshold micropulse laser, diabetic macular edema, central serous chorioretinopathy, retinal vein occlusion

## Abstract

Subthreshold micropulse laser treatment has been intensively used for selected retinal diseases in the last decade; however, the exact mechanism of the action of lasers in the subthreshold micropulse mode is not yet fully understood. This kind of treatment is safe and cheap, and contrary to classic laser photocoagulation, it leaves the retinal cells intact. A modern theory of micropulse laser interaction with retinal tissue and a possible explanation of this mechanism are presented in this review. The authors present all the relevant literature on the application of micropulse lasers in different retinal disorders. The efficacy of this treatment is analyzed on the basis of available studies and then placed in the perspective of other therapeutic methods that are used in retinal diseases.

## 1. Introduction

Lasers have been used for decades in the treatment of retinal diseases. The first retinal photocoagulation was performed in 1956 by Meyer-Schwickerath with the use of a Xenon-arc lamp of his design that was built in cooperation with the Zeiss company [1]. The following years brought the development of ophthalmic lasers and the advent of the ruby laser, argon laser, and krypton laser. Contemporary, diode, and double frequency NdYag lasers that are based on a solid state are predominantly used for the treatment of retinal diseases. When compared to the lasers built in the fifties and sixties, they are small in size and effective in power [2]. 

For years, retinal laser treatment involved the destruction of the retinal tissue. The application of laser photocoagulation (LPC) has always been a choice between the advantages of the preservation of central vision for the price of losing peripheral visual field and bearing all of the risks of photocoagulation itself. The clinical use of retinal lasers, especially in the sixties and seventies, resulted in numerous complications, such as visual retinal tissue scarring, laser scar enlargement, secondary hemorrhages, or secondary choroidal neovascularization. Those physical reactions of the retinal tissue had their reflection in functional impairment, such as visual field loss, scotomas, or even in permanent vision loss. The advent of modern lasers has diminished the role of LPC complications by more precise delivery of the power to the tissue; however, the principle of thermal destruction of the retinal cells remained. 

Therefore, there has been a constant search for laser treatment of the retina that would deliver the benefits, but not destroy cells. Micropulse and nanopulse lasers give clinicians the opportunity to treat retinal disorders without any visible damage.

## 2. Principles of Retinal Photocoagulation and Micropulse Laser Treatment

Laser photocoagulation was used for many years as an effective treatment of retinal diseases, predominantly diabetic retinopathy. The application of LPC was based on clinical research, but without the support of large randomized trials. Those were conducted as late as in the seventies and eighties by the Diabetic Retinopathy Study Group (DRS) and the Early Treatment Diabetic Retinopathy Study Group (ETDRS) [3,4]. Despite solid clinical evidence for the effectiveness of LPC, the principle of LPC action is still not well understood. 

Laser interaction with human tissue depends on the laser wavelength, pulse duration, and irradiance (energy per area) [5]. Modern retinal photocoagulators are usually Yellow 577 nm or double frequency Nd: Yag 532 lasers. Those wavelengths present the retinal tissue are predominantly absorbed by retinal pigment epithelium (RPE), melanin of the choroid, and blood. Additionally, yellow wavelengths are not absorbed by carotenoids of the macular pigment, which makes them safer in the treatment in the macular area. The main laser-tissue interaction in LPC is a thermal effect due to an increase of the retinal tissue temperature by tens of degrees Celsius. A visible result of such a laser burn is a scar on the level of RPE. The formation of a scar also involves a destruction of the photoreceptors within this area; however, the inner retina is supposed to be intact. 

There are two main theories about how laser photocoagulation can improve retinal function. The first theory concentrates on the improvement of oxygenation of the retina. Photoreceptors are cells that require vast amounts of oxygen for normal function. A destruction of some of the photoreceptors, which is a result of LPC, reduces the oxygen consumption by the retina and improves the oxygenation of its remaining part. In consequence, the production of vasoproliferative agents (mainly vascular endothelial growth factor—VEGF), which is a normal response to hypoxia, significantly diminishes. Additionally, scars are supposed to serve as bridges for oxygen passage from the choroid to the retina, thus improving its oxygenation [6]. 

The second theory puts stress on the metabolic action of retinal lasers. The insertion of laser power on the RPE not only results in tissue scarring, but also in the production of “heat shock proteins” (HSP) by surrounding the laser spot surviving tissue [7]. This stress-related response of the retinal tissue results in the immunomodulation of cell function, triggers repair processes, and then normalizes the production of certain cytokines, which reduces chronic inflammation that was previously present in the retina [8]. However, intensive photocoagulation is also a source of the acute inflammatory process itself. This might be seen in the potential complications of panretinal photocoagulation (PRP), such as a worsening of the macular edema. 

If the second theory was true, then the question of whether the destruction of retinal cells is necessary in achieving an immunomodulatory response of retinal cells had to be brought up. The downmodulation of power levels that were used for laser treatment could possibly result in a reduction of an undesirable acute inflammatory reaction and fewer complications. In other words, achieving just the positive regulatory effect of laser treatment could be the question of adjusting laser parameters and its mode of application. The concept of subthreshold and sublethal laser action has grounds in research on continuous wave lasers that are used in the treatment of diabetic macular edema (DME) [9]. Based on that research, ETDRS for years recommended the modified laser photocoagulation protocol for DME, which indicated the use of low laser power to produce only barely visible burns to the retina [10]. However, the endpoint of this therapy was limited, but still the damage to the retinal cells. 

The introduction of a micropulse mode of action of retinal lasers that were used in subthreshold power settings seems to be a consistent answer to the raised question. The micropulse mode is, in fact, an additional feature of commercially available lasers of different wavelengths: 532 nm, 577 nm, or 810 nm. The laser impact is divided into many repetitive short impulses (measured in microseconds), between which there are intervals that enable retinal tissue to cool down. The effective time of work of the laser is called a duty cycle (DC), and in retinal diseases, it is empirically set to 5% with 0.2 s. duration of the exposure envelope. Schematic explanation of the difference between the mode of action of conventional and micropulse laser is presented on Figure 1.

The idea of subthreshold laser therapy is not to leave any marks on the retina, meaning spots that could be detected by any available diagnostic tools, such as biomicroscopy, fundus autofluorescence (FAF), fundus angiography, or optical coherence tomography. Numerous studies confirm the safety of subthreshold micropulse laser treatment (SMPLT) with no detectable damage to RPE or photoreceptors [11,12,13,14]. Laser power is set at a low level, so that the laser impact does not leave any traces on the retina. In consequence, only a limited thermal impact is exerted on the tissue, without any lethal effect. The first results of SMPLT of DME were reported in 1997 by Friberg [15]; however, more extensive studies followed later [16]. In the clinical trial that was conducted by Lutrull, the treatment was described as low intensity and high density. The whole area of retinal edema was covered with confluent laser spots, without the direct treatment of the microaneurysms. The results were not worse than those achieved with classic LPC, but without any visible retinal damage. This high-density technique is typical for SMPLT. As the concept of action of SMPLT is to trigger the production of HSP by the RPE, then as many cells as possible should be stimulated. SMPLT is usually performed with multispot type of laser with spacing set to 0. 

## 3. Subthreshold Micropulse Laser Treatment and Biological Effects

As it was mentioned before, it is believed that the main mechanism of action of SMPLT is to stimulate RPE for the production of HSP and, this way, immunomodulate its metabolism and improve its function. This theory is supported by experimental research. Inagaki et al. prove the up-regulation of Hsp70 in the culture of human epithelial ARPE-19 cells after the application of SMPLT, without any thermal damage [17]. The increase in Hsp70 production was dependent on the number of impulses. 

The study of Li et al. on the RPE of mice additionally proves that SMPLT suppresses the production of neovascular promoting cytokines, such as VEGF, and then up-regulates angiogenic inhibitors, such as the pigment epithelium-derived factor (PEDF), without damaging the cells [18]. 

A recent research on human subjects shows a reduction of VEGF levels and the restoration of Muller cell (MC) function as a result of SMPLT [19]. In the study by Midena et al., Muller cell activity markers and VEGF levels were measured in the aqueous humor of patients with DME ≤ 400 μm treated by SMPLT. Additionally, the inner nuclear layer (INL) thickness, where MC bodies are located, was also measured on optical coherence tomography (OCT) scans. The results showed a reduction of VEGF levels and then suggested an improvement of Muller cell metabolism by changing biomarker levels as well as a reduction of INL thickness. 

According to the latest research on mice models, SMPLT restores the oxidant/antioxidant balance within the retinal tissue and, this way, modulates retinal cell apoptosis, counteracting the programmed cell death [20]. 

SMPLT also improves retinal perfusion in DME. Vujosevic et al. reported that SMPLT leads to a reduction of the foveal avascular zone (FAZ) area in the level of deep capillary plexus (DCP) and a reduced number of microaneurysms and area of cysts in both superficial capillary plexus (SCP) and DCP [21]. 

## 4. Laser Power Settings and Safety Issues

In evaluating the safety of laser treatment, the American National Standards Institute developed the concept of maximum permissible exposure (MPE), which was set at 1/10th of the laser exposure level that was to produce a biological effect. Most of the studies analyzing the safety of SMPLT report effective treatment between 18x and 55x MPE [12,17,22]. The safety of SMPLT was also analyzed according to the duty cycle. The previously quated study by Lutrull reports a reliable safety of 5% DC in comparison with 10% DC, where the risk of biological damage to the retina increases 10-fold [12]. 

The controversial subject is determining the power level of the SMPLT. So far, two therapeutic approaches exist: laser power titration or setting power to fixed low values (ex. 200–300 mW for the Yellow laser). Titration is recommended to be performed on the border of the edematous and a healthy retina. Power should be gradually increased until the threshold value is reached: the laser impact is barely visible as retinal whitening. For SMPLT, 30%–50% of the threshold value should be used. The titration method bears in itself the risk of overtreatment [23]. First, it is difficult to precisely determine which position on the retina is suitable for titration. Secondly, after reaching of the threshold power value, there is the question of how much to reduce the laser power—usually, authors recommend a 50% power reduction. Literature confirms SMPLT efficacy in the treatment of central serous chorioretinopathy (CSCR) or DME, also with low fixed power levels [24,25,26]. This fact could be explained by the theoretical approach. The therapeutic effect of cytokines, which are excreted as a result of SMPLT exposure, increases significantly, with only a little change in its level. It means that the therapeutic effect can be achieved after reaching a certain threshold irradiance. After that, increasing laser parameters does not enhance the biological effects but increases the risk of damaging the retina [8]. 

The safety of SMPLT has been assessed in many published studies that are listed in the Table 1, Table 2, Table 3 and Table 4. As it was mentioned before, the idea of application of SMPLT is to treat retina without any detectable damage. The lack of any trace visible in autofluorescence images and lack of discontinuity of retinal layers on the level of photoreceptors on SOCT scans is necessary in considering the safe and proper performance of SMPLT. Most of the researchers examine retina with SOCT and FAF after SMPLT and they do not report any retinal scarring. So far, very few complications of SMPLT are reported and they can be attributed to the method of setting of laser parameters [23]. However, the potential risk of the damage to retina exists and it has to be considered, especially when the central part of the fovea is treated. So far, there is no consent regarding transfoveal treatment with SMPLT. Some authors, like J. Lutrull, treat the fovea and foveola with 810 nm micropulse laser. Most of clinicians, however, avoid treating the very central part of the retina. Usually, the fovea is treated but foveola is spared. Overtreatment might result in thermal coagulation of the retinal tissue, what has to be avoided. SMPLT is used in the treatment of retinal diseases that involve the fovea, such as central serous chorioretinopathy (CSCR), diabetic macular edema, or edema secondary to retinal vein occlusion (RVO), so the potential risk of this therapy has to be addressed. When considering up to date research, titration of laser power seems to create more risk, than procedure with fixed low laser parameters. Also, it has to be stressed that most of medical equipment manufacturers decline to recommend transfoveal treatment with a micropulse laser. 

## 5. Clinical Application of SMPLT

SMPLT has been tried with different results in the treatment of the following retinal disorders:central serous chorioretinopathy;diabetic macular edema;proliferative diabetic retinopathy (PDR); andmacular edema secondary to retinal vein occlusion.

## 6. SMPLT in CSCR

CSCR is a relatively common retinal disorder that has been well described in medical literature; however, its pathogenesis is not yet fully understood [27]. Morphologically, CSCR presents an accumulation of serous fluid under the neurosensory retina. Areas of subretinal fluid are sometimes accompanied by RPE detachments (pigment epithelium detachment—PED) and alterations of the RPE (atrophy or hypertrophy), which is characteristic in the chronic form of CSCR. Contemporarily, it is believed that the source of pathology in CSCR lies in the choroid rather than in the retina [28,29,30]. 

The disease affects mainly young males between 20–50 years of age, with type A of personality, who are often exposed to prolonged stress [31]. In most cases, CSCR is presented in an acute form, which has a benign course and usually recedes spontaneously within a few months after the onset of symptoms. Visually, acuity after such an episode is not usually significantly affected; however, patients often complain about a slight decrease in the quality of vision. The chronic form of CSCR is, however, a major therapeutic problem. The prolonged presence of subretinal fluid leads to retinal thinning and the loss of photoreceptors and, in consequence, a significant deficit in visual acuity [32]. 

For many years, different therapeutic options have been sought to treat the chronic form of CSCR. Before the advent of photodynamic therapy (PDT) and SMPLT, laser photocoagulation was shown to be an effective form of treatment in selected cases of chronic CSCR [33,34,35]. Usually, green 532 nm or yellow 577 nm lasers are used for the procedure. Indications for this kind of therapy were proposed by Gass [35]. Laser photocoagulation was used in treating cases where symptoms persisted for more than four months or the recurrent form of CSCR. Fluorescein angiography was required in the planning of the treatment. It is permissible to safely treat the foci of leakage located at least 300 to 500 μm from the foveola. Laser photocoagulation is, however, not suitable for treating all cases of CSCR. There is a substantial number of cases where the leakage point is located within the center of the fovea or is itself difficult to define. Moreover, laser photocoagulation that is close to the fovea results in permanent scotoma, which may cause the patient a degree of discomfort. That is why, at present, other forms of treatment are preferable. 

One of them—well established and examined—is photodynamic therapy [36,37,38]. The procedure involves long infrared laser exposure of the area of leakage after the administration of an intravenous photosensitizing substance, in this case, verteporfin. In CSCR, the verteporfin dose is half of the dose that is normally used for the treatment of an exudative form of age-related macular degeneration (AMD). Irradiance is also reduced to the level of half-fluence in comparison to AMD treatment. 

Another modern method of treatment of chronic CSCR is SMPLT, intensively used since 2008. It is interesting to link SMPLT efficacy in CSCR to the possible mechanism of action of a subthreshold micropulse laser. Although it is believed that in CSCR pathology lies in choroid, rather than in the RPE, it is the retinal pigment epithelium that transfers SRF to choroidal vessels. SMPLT improves cell function, thus improving the pumping efficacy of RPE. Besides, the production of cytokines after stimulation by SMPLT also probably reduces inflammatory processes accompanying that disease. 

Table 1 lists the results of major studies employing SMPLT for the treatment of CSCR. 

In all of the studies except two, CSCR was treated if the disease lasted longer than three months, which meets criteria of CSCR chronicity that is generally accepted in medical literature. One study treated patients with a duration of symptoms longer than six weeks (Scholz et al. 2015) and one treated both acute and chronic cases (Lutrull 2016).

Most of the studies reveal a high efficacy of SMPLT in improving retinal morphology in chronic CSCR. However, according to this data, a complete resolution of SRF is achieved in 60%–80% of cases. What is more, BCVA gain after treatment is usually not significant. In most of the studies, BCVA improvement is no better than 1 line on the Snellen chart. This poor functional effect is also noted after PDT treatment. Table 2 lists studies comparing SMPLT and PDT in the treatment of chronic CSCR. 

As can be seen, only the last trial (PLACE trial by van Dijk et al.) places PDT above SMPLT as the more effective therapy in providing the resolution of SRF. This result is not confirmed by any other studies and it might be attributed to the methodology of SMPLT that was used in this particular project. The authors applied spots of micropulse laser just to the areas of choroidal leakage basing on indocyanine green angiography (ICGA). Other clinicians usually use OCT for SMPLT guidance and cover all of the areas of subretinal fluid presence with the confluent laser foci.

In most of the published results, both PDT and SMPLT are similarly effective in treating chronic CSCR, although for the patient, SMPLT is definitely cheaper and less troublesome. Taking that into consideration, SMPLT could be used as the first line of treatment, leaving PDT as the second choice for non-responders. 

Other methods, such as intravitreal injection of anti-VEGF medications appeared to be unsuccessful in treating pure serous chorioretinopathy [37,55]. Oral mineralocorticoid pathway inhibitors are used as a support of invasive therapies; however, so far, there has been no consent regarding their efficacy in the treatment of chronic CSCR [27,56,57,58]. 

If the functional results of the treatment of chronic CSCR are so poor, then the question of employment of SMPLT in the acute form of CSCR has to be raised. Up until recently, only chronic forms of CSCR were treated. Usually, clinicians were delaying any invasive treatment for 3–6 months for a spontaneous remission of symptoms. However, with the advent of SMPLT, ophthalmologists have a safe, non-damaging tool to the retina for the treatment of CSCR. Arora et al. compared the results of early treatment of CSCR with observation [58]. Patients who undergone SMPLT in the acute phase of CSCR had better final BCVA and better contrast sensitivity than the patients that were just observed. 

## 7. SMPLT Treatment of DME

Treatment of diabetic macular edema has evolved in the last few decades. For many years, LPC was, in fact, the only effective treatment of DME that preserved the actual visual acuity [59]. This kind of therapy, however, involved the destruction of photoreceptors and it thus had serious side effects, such as visual field scotomas and the possibility of secondary choroidal neovascularization [60]. That is why, nowadays, LPC for the treatment of DME is used very rarely, usually when other therapeutic methods are not available or are contraindicated. SMPLT as a non-damaging retinal therapy has been used for the treatment of DME for many years now. The possible and suggested mechanism of the action of SMPLT in DME is the production of cytokines that work similarly to antiVEGF medications. Stimulation of retina by SMPLT reduces VEGF levels, which has been proved on an animal model and in humans. In consequence, the most potent pro-edema factor is compromised. Additionally, other cytokines perform anti-inflammatory action, which in summary reduces vessels permeability. As RPE cell function is improved, the elimination of intraretinal and subretinal fluid is also more efficient. 

Table 3 lists the results of major studies on its efficacy in treating that disease. 

As can be seen, the results of the treatment are good, especially when we compare SMPLT with classic LPC—the superiority of SMPLT over LPC has been proven in a few studies [25,26,66,67,74]. However, it has to be stressed that a morphological improvement is often better than a functional one. More than one ETDRS or the Snellen line rarely improved the BCVA. Scholz et al. have calculated the average improvement in the treatment of DME by SMPLT in the material from 11 studies (613 patients) [75]. The average BCVA change was +1.26 ETDRS letters (range −6.6 to 19) and the average CRT reduction was −74.9 µm (range −138 µm to 48 µm). 

Contemporarily, the most successful treatments for central DME are intravitreal injections of anti-VEGF medications and steroids, with the latter being recommended as the second line of treatment [76]. Therapy with intravitreal injections is obviously invasive, but also non-damaging for the retina. The visual acuity gain after anti-VEGF therapy in DME is generally superior to SMPLT [77]. A large Diabetic Research Clinical Research Network (DRCRnet) study reports the mean BCVA gain at two years of 12.8 letters for aflibercept, 10.0 letters for bevacizumab, and 12.3 letters for ranibizumab. In patients with a worse baseline BCVA (20/50 to 20/320), the improvement is even better, at 18.1, 13.3, and 16.1 letters, respectively [78]. A properly conducted intravitreal therapy, however, requires strict compliance and involves numerous invasive procedures, which intravitreal injections definitely are. Besides, when choosing a suitable therapy for DME, the economic aspect should also be taken into consideration. Treatment with intravitreal injections is expensive. SMPLT is much cheaper, and thus it would be economically rational to include it in the process of treating DME. 

Moisseiev et al. proves that combined therapy—anti-VEGF (in this case, ranibizumab) plus SMPLT—significantly reduces the burden of intravitreal injections [79]. Patients who were subject to such a therapy required significantly fewer injections than those that were just treated with ranibizumab alone (2.6 versus 9.3 at the end of the follow-up). 

The European retinal society “Euretina” accepts the use of SMPLT for DME as an option in cases with early diffuse retinal edema and good visual acuity [76]. Generally, there is a tendency for the of SMPLT in small or moderate DME. Studies prove that initial CRT has an important impact on the response to SMPLT [72,80]. SMPLT is rarely effective in treating edema larger than 400 µm. 

An interesting subject is also the adjustment of laser settings for the treatment of DME. Generally, 5% DC is preferred by most of the authors; however, there are reports regarding the better results that were achieved with 15% DC [81]. 

Therefore, it seems that, in some selected cases, there may be a place for the SMPLT in central DME. These cases include patients that are reluctant to receive intravitreal treatment, non-compliant patients, patients in whom antiVEGF therapy is contraindicated due to systemic reasons, patients who can not bare financial burden of intravitreal therapy, and a group of patients with small diabetic edema of short duration and good visual acuity. Further research is needed to precisely indicate the position of SMPLT in the treatment of DME, in particular, clinical trials comparing anti-VEGF therapy and SMPLT in the eyes, with relatively small central DME carrying good visual acuity.

## 8. SMPLT for Severe NPDR or Proliferative Diabetic Retinopathy

Classic panretinal photocoagulation (PRP) is a standard for treating severe NPDR or proliferative diabetic retinopathy. The efficacy of such an approach is confirmed in milestone DRS and ETDRS studies [4,82]. PRP always results in peripheral visual field defects or general visual field narrowing, which could be troublesome, especially for younger patients. Therefore, other therapeutic methods were sought that would spare peripheral retina. AntiVEGF therapy has been tried as an alternative to PRP with functional success; however, it requires regular intravitreal injections and numerous follow-up visits, which is difficult to comply with for some patients [83,84]. Subthreshold micropulse panretinal photocoagulation has also been tested as an alternative to classic PRP, however, only in a couple of trials. Lutrull et al. published the first results in 2008 [22]. Authors observed 99 eyes of patients with severe NPDR or proliferative diabetic retinopathy (PDR), who were subject to SMPLT PRP. After 12 months, the risk for progression to vitreal haemorrhage was assessed at 12.5% and risk for pars plana vitrectomy at 14.6 %. A second study by Jhingan et al. included 10 patients (20 eyes) with severe NPDR or low-risk PDR [85]. Each eye was randomized to receive either classic PRP or SMPLT PRP. At nine months, only one eye in SMPLT group progressed to vitreal haemorrhage and then required classic PRP. Eyes from classic PRP group revealed worse retinal sensitivity when compared to eyes that received SMPLT PRP. 

Theoretically, the metabolic action of SMPLT should improve the function of the RPE cells, thus improving retinal oxygenation and stimulating the production of substances acting against vascular growth factors. In a sense and within reasonable proportions, SMPLT PRP should work similarly to antiVEGF injections. So far, the results are promising: SMPLT PRP appeared as not inferior to classic PRP, but preserving retinal sensitivity. At this point, however, this kind of treatment has to be considered as experimental. The question to be answered is the schedule of the follow-up of the patient after SMPLT PRP and the frequency of procedures that are needed to prevent the development of retinal neovascularization. Besides, it needs to be said that full SMPLT PRP with confluent laser spots would take a lot of time and it would have to be conducted during a few laser sessions. 

## 9. SMPLT in ME Secondary to RVO

Retinal vein occlusion is a common retinal disease that could result in peripheral ischemia and macular edema. For many years, LPC has been used as a treatment for those clinical entities [86,87]. Before the advent of antiVEGF injections and SMPLT, LPC was, in fact, the only approach that appeared to be effective, especially in ME secondary to branch retinal vein occlusion (BRVO). Again, as in the treatment of DME, LPC in ME secondary to RVO improves the retinal morphology and function for the price of the destruction of photoreceptors.

That is why the present ophthalmological standard recommends intravitreal antiVEGF or steroids in the treatment of macular edema secondary to RVO [88,89]. However, the functional results of such a treatment are not always satisfactory. Besides, most of the trials refer to the efficacy of intravitreal therapies to LPC, and not to SMPLT [90,91,92]. The results of those studies definitely favor intravitreal therapies against LPC. 

Nevertheless, SMPLT was tried as an alternative to LPC and later on to antiVEGF, although studies on that subject are rather scarce. 

The results of SMPLT treatment of macular edema secondary to branch retinal vein occlusion are presented in Table 4. 

In theory, SMPLT should reduce inflammation and improve the elimination of fluid due to RPE stimulation (production of cytokines and boosting pumping activity). In RVO, inflammatory processes and vascular hyperpermeability are probably more intense in comparison to other retinal vascular diseases. That could explain why results of SMPLT of macular edema secondary to RVO are not always satisfactory. In these cases, SMPLT is probably not potent enough to overweigh the benefits of intravitreal steroid or antiVEGF treatment. 

Still, there are some inconsistencies in the results that are presented in Table 4. SMPLT seems to be superior to classic LPC in improving BCVA, especially as it leaves retinal cells intact. However, a comparison of the efficacy of SMPLT and intravitreal therapies in the treatment of ME that is associated with RVO does not unequivocally favor any of those therapies. Studies by Parodi show significantly better results with the use of intravitreal triamcinolone or bevacizumab than SMPLT alone [94,96]. On the other hand, the latest research does not favor intravitreal ranibizumab against SMPLT in the treatment of ME secondary to RVO [97,98]. A number of available clinical trials also have to be taken into consideration. 

Further research is definitely needed to find the place of SMPLT in the treatment of ME in RVO. Nevertheless, patients with small macular edema or patients that were disqualified from intravitreal therapies due to systemic conditions or financial reasons could still benefit from SMPLT. 

## 10. Conclusions

SMPLT emerges as a new non-invasive and effective treatment in selected cases of retinal diseases. Therapy by subthreshold micropulse laser is cheap and safe and it is not unpleasant for the patient. It has high efficacy in resolving subretinal fluid in CSCR, and in this particular disorder, it could be treated as the first line of treatment. The question of timing its application is still to be discussed; however, there is some evidence in support of its use in the active form of the disease, without waiting for spontaneous remission. 

The position of SMPLT in the treatment of DME and ME secondary to RVO is yet to be determined. Intravitreal therapies with antiVEGF or steroid medications are therapies of choice in those disorders; however, SMPLT can be used with success in some cases, especially when the use of the abovementioned drugs is contraindicated. 

Panretinal photocoagulation with SMPLT is currently a form of experimental therapy and its efficacy needs to be confirmed in further research.

## Figures and Tables

**Figure 1 jcm-08-00242-f001:**
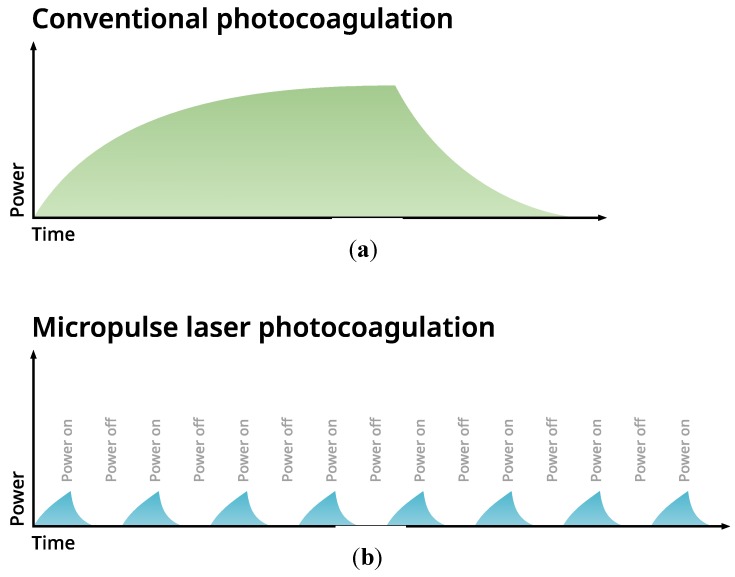
(**a**,**b**) Mechanism of action of continuous wave laser versus subthreshold micropulse laser treatment (SMPLT). (**a**) In application of continous wave laser the power is distributed evenly within the duration of an impact. (**b**) In SMPLT, laser impact is delivered in a train of short impulses between which there are intervals that enable retinal tissue to cool down.

**Table 1 jcm-08-00242-t001:** Efficacy of SMPLT in the treatment of CSCR in major clinical trials. CSCR: central serous chorioretinopathy, CRT: central retinal thickness, SRF: subretinal fluid, BCVA: best corrected visual acuity, FA: fluorescein angiography, ETDRS: Early Treatment Diabetic Retinopathy Study.

Study	Duration of CSCR	No of Eyes	Morphological Outcome	Functional Outcome
Chen et al. 2008 [39]	Longer than 4 months	26	Reduction of SRF in 100% Total resorption of SRF in 73% An average reduction of CRT by 148 μm to 203 μm depending on the subgroup of patients	A statistically significant improvement of BCVA in 100% of patients BCVA gain by at least 3 lines in 57.7%, a gain between of 1–3 lines in 23.1%
Lanzetta et al. 2008 [40]	Longer than 3 months	24	A reduction of SRF in 75% Total resorption of SRF in 71% An average reduction of CRT from 328 µm to 168 µm	An improvement of BCVA from 20/32 to 20/25
Koss et al. 2012 [41]	Longer than 3 months	16	An average reduction of CRT from 419 μm to 325 μm; a lack of leakage on FA in 87.5%	An average improvement of BCVA: 6 letters ETDRS
Roismann et al. [42]	6 months or longer	10	CRT reduction from 420 μm to 265μm	A significant increase in BCVA after 3 months from 35.4 to 47.9 letters ETDRS
Malik et al. 2015 [43]	3 months or longer	11	A reduction of SRF in 72% Mean CRT reduction: 97 μm	An average increase in BCVA: 6 letters ETDRS
Abd Elhamid et al. 2015 [44]	3 months or longer	15	A reduction of SRF in 100% Total resorption of SRF: 73% after 3 months and 86% after 6 months	An improvement of BCVA from 0.67 Snellen to 0.85 Snellen
Yadav et al. 2015 [13]	Longer than 3 months	15	A reduction of SRF in 100% Total resorption of SRF: 40%	An improvement of BCVA from 20/40 Snellen to 20/30 Snellen, statistically significant
Kim et al. 2015 [45]	6 months or longer	10	CRT reduction from 349.2 μm to 261.2 μm	BCVA improvement from 0.21 logMar to 0.035 logMar
Scholz et al. 2015 [46]	Longer than 6 weeks	38	A reduction of SRF in 74% An average reduction of CRT by 115 μm	BCVA improvement by 0.06 logMAR
Lutrull JK. 2016 [47]	1–7 months, 3.6 on average	11	CRT reduction from mean 508 µm to mean 250 µm	BCVA improvement from mean 20/37 to mean 20/24
Ambiya V et al. 2016 [48]	3 months or more	10	SRF reduction in 100 %; complete SRF resorption in 60% at 6 months; CRT reduction from average 298 µm to 215 µm	BCVA change from 73.3 letters ETDRS to 76.9 letters ETDRS at 6 months, not significant
Maruko et al. 2017 [49]	Longer than 3 months; patients with focal leakage on FA only	14	Total resorption of SRF in 64.3% An average CRT reduction from 328 μm to 192 μm	BCVA change from 0.96 to 0.94 Snellen, statistically insignificant
Gawęcki et al. 2017 [24]	Longer than 4 months	51	Total resorption of SRF in 70.6% An average reduction of CRT from 337.6 μm to 260µm	An average improvement of BCVA +0.08 logMAR, statistically significant
Arsan et al. 2018 [50]	Longer than 3 months	39	Median follow-up: 17.82 months (13–24 months); An average CRT reduction from 369 μm to 250 µm	An improvement of BCVA in 89.7%

**Table 2 jcm-08-00242-t002:** Studies comparing the results of treatment by SMPLT or photodynamic therapy (PDT) in chronic CSCR. CSCR: central serous chorioretinopathy, CRT: central retinal thickness, SRF: subretinal fluid, BCVA: best corrected visual acuity, FA: fluorescein angiography, ETDRS: Early Treatment Diabetic Retinopathy Study SMPLT: subthreshold micropulse laser treatment PDT: photodynamic therapy.

Study	Duration of CSCR	Results in SMPLT Group	Results in PDT Group
Krets et al. 2015 [51]	3 months or longer	20 cases; a reduction of leakage on FA in 60%; CRT change after 4 months: −69.7 μm; BCVA change +6.7 ETDRS letters	24 cases; half dose PDT; a reduction of leakage on FA in 67 %; CRT change after 4 months: −109.8 μm; BCVA change +8.5 ETDRS letters
Scholz P et al. 2016 [52]	6 weeks or longer	48 cases; SRF reduction in 79%; CRT change from mean 445 μm to mean 297 μm; BCVA improvement from 0.39 logMAR to 0.31 logMAR	52 cases; half dose PDT; SRF reduction in 59%; CRT reduction from mean 398 μm to mean 322 μm; BCVA improvement from 0.35 logMAR to 0.31 logMAR
Özmert et al. 2016 [53]	Longer than 6 months	15 cases; SRF reduction in 87%, complete resolution in 80%; CRT reduction from average 287.3 μm to 138.0 μm; BCVA improvement from 67.3 letters ETDRS to 71.5 letters ETDRS (not significant)	18 cases; low-fluence PDT; SRF reduction in 78%, complete resolution in 72%; CRT reduction from average 242.8 μm to 156.9 μm; BCVA improvement from 60.7 letters ETDRS to 64.5 letters ETDRS (not significant).
Van Dijk EHC et al. (PLACE trial) 2018 [54]	6 weeks or longer	90 cases; complete resolution of SRF in 28.8 % after 7–8 months: BCVA improvement 1.39 ETDRS letters after 6–8 weeks and 4.48 letters after 7–8 months	89 cases; complete resolution of SRF in 67.2% after 7–8 months; BCVA improvement of 4.6 ETDRS letters after 6–8 weeks and r 6.78 letters after 7–8 months.

**Table 3 jcm-08-00242-t003:** Major studies on SMPLT treatment of DME. LPC: laser photocoagulation, CSME: clinically significant macular edema, CRT: central retinal thickness, SRF: subretinal fluid, BCVA: best corrected visual acuity, DME: diabetic macular edema, NPDR: non-proliferative diabetic retinopathy, FA: fluorescein angiography, ETDRS: Early Treatment Diabetic Retinopathy Study, MV: macular volume, ND: normal density, HD: high density, mfERG: multifocal electroretinography, FAF: fundus autofluorescence.

Study	No of Eyes	Design of the Study	Results
Laursen et al. 2004 [61]	SMPLT 12 eyes LPC 11 eyes	A comparison of the efficacy of SMPLT versus LPC in the treatment of CSME. An evaluation of BCVA and CRT (OCT).	A significant reduction of CRT for focal edema in the SMPLT group after 3 months. BCVA stable in both groups.
Luttrull et al. 2005 [16]	SMPLT 95 eyes with CSME	The results of SMPLT (BCVA, ME status) in CSME in patients with mild and moderate NPDR	A significant improvement of BCVA in 85%. A reduction of CSME in 96%, a resolution of CSME in 79%.
Lutrull et al. 2006 [62]	18 eyes with CSME	A reduction of CRT in OCT after SMPLT at 3 months	A mean reduction of CRT by 24%. 11 eyes with recurrent or persistent CSME gained a mean CRT reduction of 59%
Sivaprasad et al. 2007 [63]	25 eyes with CSME	BCVA and FA status after 3 years	An improvement of BCVA in 84% after 1st year sustained in 92% after 3rd year. CSME decreased in 92% after 1st year and completely regressed in 88% after 1st year. Recurrence of CSME in 28% after the 3rd year.
Nakamura et al. 2010 [64]	28 eyes with DME	BCVA, retinal sensitivity in microperimetry and CRT after 3 months	A significant reduction of CRT on average from 481 µm to 388 µm. A significant improvement of BCVA from 0.47 logMAR to 0.4 logMAR. No significant change in retinal sensitivity.
Ohkoshi et al. 2010 [65]	43 eyes with CSME ≤600 µm	BCVA, CRT and MV after 3 months	A significant reduction in CRT from 341.8 µm to 300.7 µm. No significant change in BCVA and MV.
Vujosevic et al. 2010 [25]	SMPLT 32 eyes with CSME LPC 30 eyes with CSME	A comparison of SMPLT and LPC group according to BCVA, CRT and retinal sensitivity in microperimetry at 12 months	BCVA stable in both groups. A reduction of CRT significant in both groups with no difference between the groups. Retinal sensitivity improvement in SMPLT group and a decrease in LPC group.
Lavinsky et al. 2011 [66]	ND SMPLT 39 eyes, HD SMPLT 42 eyes, LPC 42 eyes Treatment naive cases with DME and BCVA between 20/40 and 20/400	A comparison of BCVA and CRT at 12 months.	Biggest improvement of BCVA in HD SMPLT: 0.25 logMAR versus w 0.08 logMAR in LPC group and not significant w 0.03 logMAR in ND SMPLT group. A reduction of CRT in all groups, greatest in HD SMPLT (154 µm) but not statistically different from 126 µm in the LPC group.
Venkatesh et al. 2011 [67]	SMPLT 23 eyes LPC 23 eyes with CSME	CRT, mfERG, BCVA and contrast sensitivity at 6 months	A decrease of CRT, an improvement of BCVA and contrast sensitivity similar in both groups. Better mfERG results in SMPLT group (fewer areas of signal void).
Takatsuna et al. 2011 [68]	56 eyes with DME	BCVA and CRT at 12 months	A reduction of CRT from 504 µm to 320 µm. No statistical improvement in mean BCVA. An improvement of > 0.2 logMAR units in 17.8% of cases.
Inagaki K et al. 2012 [69]	21 eyes with DME	Treatment with SMPLT plus direct LPC. BCVA and CRT at 6 months.	A significant reduction of CRT. No change in mean BCVA.
Othman et al. 2014 [70]	220 eyes with CSME and no ischemia. Group 1: 187 eyes—primary treatment by SMPLT Group 2: 33 eyes—SMPLT as secondary treatment after LPC	DC 15%. Follow up 12–19 months. An evaluation of BCVA and CRT.	A significant improvement of BCVA in group 1 at 4 months and stable after (0.21 logMAR to 0.18 logMAR). Stable BCVA without a significant improvement in group 2. A significant reduction of CRT in both groups.
Luttrull et al. 2014 [71]	39 eyes with CSME and BCVA of 20/40 or better	CRT and BCVA at 4–7 months of follow up.	An improvement of BCVA by an average of 0.03 logMAR units. A significant decrease in CRT.
Mansouri et al. 2014 [72]	63 eyes with DME	A comparison of results (BCVA, CRT) in group 1 with ME ≤400 µm (33 eyes) and group 2 with ME > 400 µm (30 eyes) Follow up 6 months	Group 1: a significant improvement of BCVA by average 0.2 logMAR and a significant reduction of CRT by an average of 55 µm. Group 2: no significant change in BCVA or CRT.
Inagaki et al. 2015 [73]	53 eyes with DME 24 eyes—810 nm SMPLT 29 eyes—577 nm SMPLT	A comparison of the effects of treatment (BCVA and CRT) of 577 nm and 810 nm SMPLT combined with direct LPC of microaneurysms	A significant reduction of CRT in both groups without a difference between the groups. BCVA stable in both groups.
Vujosevic et al. 2015 [12]	53 eyes with CSME <400 µm. 26 eyes—577 nm SMPLT 27 eyes 810 nm SMPLT	A comparison of the effects of SMPLT between 577 nm and 810 nm groups at 6 months. An evaluation of CRT, MV, choroidal thickness, BCVA, retinal sensitivity. Safety: FAF, FA, the integrity of outer retinal layers.	No significant differences in any parameters between the groups. A significant improvement of retinal sensitivity in both groups. No visible scars on FA, FAF and OCT.
Fazel et al. 2016 [74]	68 eyes with CSME < 450 µm. SMPLT 34 eyes LPC 34 eyes	A comparison of SMPLT and LPC (BCVA, CRT, MV) at 4 months.	A significant reduction of CRT in both groups (357.3 µm to 344.3 µm in SMPLT group and from 354.8 µm to 349.8 µm in LPC group). BCVA and MV improved significantly in SMPLT group only (BCVA from 0.59 logMAR to 0.52 logMAR)

**Table 4 jcm-08-00242-t004:** Results of SMPLT treatment of macular edema (ME) secondary to retinal vein occlusion. SMPLT: subthreshold micropulse laser treatment, CME: cystoid macular edema, IVT: intravitreal triamcinolone, IVB: intravitreal bevacizumab, IVR: intravitreal ranibizumab, LPC: laser photocoagulation, ME: macular edema, CRT: central retinal thickness, BCVA: best corrected visual acuity, MV: macular volume, BRVO: branch retinal vein occlusion.

Study	No of Eyes	Study Design	Results
Parodi et al. 2006 [93]	36 eyes 17 eyes—SMPLT 19 eyes—GRID LPC	A comparison of the results of treatment of ME by SMPLT and LPC at 24 months	No difference between the groups in BCVA, CRT, MV at 12 months. At 24 months, a gain of at least 10 letters was noted in 65% of patients after SMPLT and 58% of patients after LPC.
Parodi et al. 2008 [94]	24 eyes: 13—SMPLT alone 11—SMPLT plus IVT	A comparison of the results of treatment of SMPLT alone and SMPLT combined with IVT at 12 months	A gain of at least 10 letters at 12 months: 91% in SMPLT plus IVT group 62% in SMPLT alone group. Statistically significant difference.
Inagaki et al. 2014 [95]	32 eyes with longstanding ME after BRVO) (at least 6 months) Group 1: BCVA ≤ 20/40. Group 2: BCVA > 20/40.	A comparison of the results of SMPLT in both groups at 6 and 12 months.	A significant reduction of CRT in both groups, no difference between the groups. No significant improvement of BCVA in both groups.
Parodi et al. 2015 [96]	35 eyes with ME secondary to BRVO after previous LPC Group 1 (18 eyes)—SMPLT. Group 2 (17 eyes)—IVB.	A comparison of the efficacy of IVB versus SMPLT in recurrent ME at 12 months.	A reduction of CRT in the IVB group only: from 484 µm to 271 µm. BCVA improvement in IVB group only: from 0.94 logMAR to 0.72 logMAR. A gain of at least 3 lines in 58% of patients. No significant effect for CRT or BCVA in SMPLT group.
Terashima et al. 2018 [97]	46 eyes with ME due to BRVO, treatment naive 22 eyes: IVR plus SMPLT 24 eyes: IVR alone	A comparison of combined therapy IVR plus SMPLT versus IVR alone. An evaluation of BCVA, CRT at 6 months.	A significant improvement of BCVA and CME in both groups, without a difference between the groups. A number of injections in the IVR group were statistically higher than in the IVR plus SMPLT group (2.3 versus 1.9)
Buyru Ozkurt et al. 2018 [98]	51 eyes with ME after BRVO 27 eyes IVR group 24 eyes SMPLT group	A comparison of the efficacy of IVR versus SMPLT treatment after 12 months. An evaluation of BCVA, CRT and the number of treatments.	No difference in the final BCVA or CRT between the groups. The number of treatments in the IVR group was 3.81 and 1.5 in SMPLT group.

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
