# Peer review of "Micropulse Laser Treatment of Retinal Diseases"

_jcm, 2019, doi:10.3390/jcm8020242_

Round 1
Reviewer 1 Report
This is an interesting article. I only have a problem in the organization of the article. This article would have been more attractive if there are a lesser number of tables being used. The information displayed in the tables can be communicated through text. On the other hand, one figure for the mechanism of treatment would have been increased the impact of the article.
- use of micropulse laser treatment for retinal diseases. This is absolutely interesting for the readers who are working in the field of retinal or per se ocular diseases. There are not many papers written on the same topic except three recent reviews (Scholz P. et al, 2017; Chehade L et al. Clin Exp Ophthalmol., 2016; Shah S et al. Ophthalmic Surg Lasers Imaging Retina., 2016). All of these papers serve the criteria for one or other but is not a concise description of the whole scenario. Thus, this paper definitely will contribute significantly to the field of research. - The paper is well-written and I do not see any major typos or grammatical mistakes which need to be corrected. The only way I think the author can improve the quality of the manuscript is to include a figure. Instead of describing 'principles of retinal photocoagulation in words, the author can describe it though a diagram or a scheme. I also do not prefer the way the tables are presented. Instead of the major study, there should be description of the treatment procedures. Otherwise, it looks like a bit more repetitive (e.g., I can see 3 months so many times in the table 1). - The conclusion is well synchronized with the argument posed for the role of micropulse laser treatment in retinal diseases. But, the presentation of the paper can be modified with a bit of 'catchy' terms. Then, it will be more attractive to the readers.
Author Response
1. Thank you for the suggestion concerning the figure. I included the figure explaining the difference between the action of continuous wave laser and micropulse laser.
2. I reduced the number of tables (I removed one). However, if you agreed, I would leave the list of the studies which apply to each of the diseases. This way a reader could have easier access to the research. If you insist on removing the tables I will do that.
3. According to your suggestion I added the description of the laser treatment procedures in CSCR explaining the criteria of chronicity. Thank you for this remark.
Reviewer 2 Report
This manuscript reviewed the efficacy of subthreshold micropulse laser (SMPL) in treatment of retinal diseases, such as central serous chorioretinopathy, diabetic retinopathy and macular edema secondary to retinal vein occlusion. This review article discussed the possible mechanism of the SMPL in treatment of these disease.
Specific Comments:
1. Please discuss the possible different mechanism when SMPL used in treatment of different retinal diseases.
2. Please discuss the safety of the SMPL in treatment of different diseases.
3. PDT and anti-VEGF therapy are very commonly used in treatment of retinal diseases recently. Please discuss what kinds of patients can get benefit from SMPL treatment, if PDT and anti-VEGF therapy was not selected?
4. Reference [33] and [57] were not cited in the manuscript.
5. Page 14, Line 452. Please correct the format of citation.
Author Response
1. I added the discussion on possible mode of action of micropulse laser in different retinal diseases in each paragraph referred to each clinical entity. Thank you for that remark.
2. I added text on safety of SMPLT procedures in paragraph 4. I changed the title of the paragraph to: Laser parameters and safety issues.
3. I discussed benefits of micropulse laser treatment in non-responders to anti-VEGF or PDT treatments in each paragraph devoted to each clinical entity treated by SMPLT. Thank you for that valuable suggestion.
4. I cited references 33 and 57 in the text.
5. Page 14 line 452 – I did amend the format of citation.
Round 2
Reviewer 2 Report
This revised article is well written and the author answered all of my questions. Thanks.